# Degradation of Human Serum Albumin by Infrared Free Electron Laser Enhanced by Inclusion of a Salen-Type Schiff Base Zn (II) Complex

**DOI:** 10.3390/ijms21030874

**Published:** 2020-01-29

**Authors:** Yuika Onami, Takayasu Kawasaki, Hiroki Aizawa, Tomoyuki Haraguchi, Takashiro Akitsu, Koichi Tsukiyama, Mauricio A. Palafox

**Affiliations:** 1Department of Chemistry, Faculty of Science, Tokyo University of Science, 1-3 Kagurazaka, Shinjuku-ku, Tokyo 162-8601, Japan; 1319526@ed.tus.ac.jp (Y.O.); 1317601@ed.tus.ac.jp (H.A.); haraguchi@rs.tus.ac.jp (T.H.); tsuki@rs.kagu.tus.ac.jp (K.T.); 2FEL-TUS, Tokyo University of Science, 2641 Yamazaki, Noda, Chiba 278-8510, Japan; kawasaki@rs.tus.ac.jp; 3Departamento de Química-Física, Facultad de Ciencias Químicas, Universidad Complutense de Madrid, 28040 Madrid, Spain; alcolea@quim.ucm.es

**Keywords:** IR-FEL, human serum albumin, Schiff base, Zn(II) complex, TD-DFT, fluorescence

## Abstract

A salen-type Schiff base Zn(II) complex included in human serum albumin (HSA) protein was examined by UV-Vis, circular dichroism (CD), and fluorescence (PL) spectra. The formation of the composite material was also estimated by a GOLD program of ligand–protein docking simulation. A composite cast film of HSA and Zn(II) complex was prepared, and the effects of the docking of the metal complex on the degradation of protein molecules by mid-infrared free electron laser (IR-FEL) were investigated. The optimum wavelengths of IR-FEL irradiation to be used were based on experimental FT-IR spectra and vibrational analysis. Using TD-DFT results with 6-31G(d,p) and B3LYP, the IR spectrum of Zn(II) complex could be reasonably assigned. The respective wavelengths were 1652 cm^−1^ (HSA amide I), 1537 cm^−1^ (HSA amide II), and 1622 cm^−1^ (Zn(II) complex C=N). Degradation of HSA based on FT-IR microscope (IRM) analysis and protein secondary structure analysis program (IR-SSE) revealed that the composite material was degraded more than pure HSA or Zn(II) complex; the inclusion of Zn(II) complex enhanced destabilization of folding of HSA.

## 1. Introduction

In recent years, researches on artificial metalloenzymes, namely, docking metal complexes to proteins, such as bio-inspired catalysts, drug discovery, and biofuel cells have attracted increased attention [1,2,3,4,5,6,7,8,9,10,11,12,13,14]. Because certain proteins can deliver drugs to anticancer cells, experiments to dock with nucleic acids, as well as proteins, have also been widely performed. Metal complexes, on the other hand, exhibit redox properties using metal ions, as well as various organic ligands. In addition, since typical Schiff base complexes are relatively simple in structure and easy to synthesize, they can be applied in various fields. In this way, the potential of materials with metal complexes docked to proteins is expanding.

Tokyo University of Science possesses an infrared free electron laser (IR-FEL) oscillation facility, called “FEL-TUS”. It is a synchrotron radiation-based coherent laser that has an adjustable wavelength in the mid-infrared region (5 to 10 µm) and oscillates with picosecond pulses. Recently, FEL-TUS has succeeded in dissociating several protein aggregates [15,16,17], for example, amyloid fibrils causing Alzheimer’s disease [18,19], by excitation of amide bonds at amide I band (c.a. 6 μm). Although IR-FEL has been well used to ablate biological tissues in medicine, molecular interaction of proteins with intense infrared radiation has been less studied [20,21,22] contrary to UV light [23,24].

There are few studies on infrared laser irradiation for metal complex–protein hybrid materials [25,26,27,28,29,30], and thus this research can be regarded as a newly explored field. Absorption of IR light may suggest how to dock metal complexes into HSA and how it influences dissociation by IR-FEL [31]. Certain Cu(II) or Zn(II) complexes incorporating amino acids with phenyl moieties were prepared for photochemical and redox studies. Previously, we also prepared hybrid materials of proteins (e.g., HSA) containing metal complexes [32,33]. This provides additional properties due to metal complexes (e.g., light and redox reactions [34,35,36,37,38]). 

In this study, we synthesized a known Schiff base Zn(II) complex (**ZnL**) and HSA+**ZnL** hybrid material (Scheme 1) to compare pure HSA against radiation damage by IR-FEL [10], and performed an analysis by FT-IR infrared microscopy (IRM) (Scheme 1). Based on these spectral results, protein secondary structure analysis was carried out using IR-SSE [11]. A structural change of HSA was observed over the time of IR-FEL irradiation. In order to select the IR wavenumber of the irradiation and other experimental conditions, the experimental IR spectrum can be employed to obtain the absorption wavenumber of both HSA and **ZnL**. Furthermore, the DFT computational method was also utilized for supporting reliable assignment and providing an optimized structure of **ZnL**, considering a docking feature into the HSA molecule.

The explanation of each experiment is as follows: In Section 2.1, optical, chiroptical, and fluorescence spectra of **ZnL** and **HSA+ZnL** exhibit docking of **ZnL** into **HSA**. Especially, appropriate wavelengths of optical and fluorescence spectra leads to Fluorescence Resonance Energy Transfer (FRET), which suggests short distance approaching of the guest **ZnL** molecules. In Section 2.2, experimental and simulated IR spectra indicates the wavelength of infrared light which can be absorbed by **ZnL** molecules to protect HSA molecles damaged directly. In Section 2.3, finally, are the results of IR-FEL irradiation and changes of secondary structures of **HSA** molecules damaged, analysed from the IR spectra. The important results and discussion to interpret this study are presented in the next section.

## 2. Results and Discussion

### 2.1. Spectral Changes by Docking of Zn(II) Complex and HSA

#### 2.1.1. UV-Vis Spectra

In the first step, optical, chiroptical, and fluorescence spectra of **ZnL** and **HSA+ZnL** exhibit docking of **ZnL** into **HSA**. Figure 1 depicts UV-Vis spectra (in dimethylsulfoxide—DMSO) of HSA+**ZnL**, HSA, and **ZnL**. Because of d^10^ electronic configuration, **ZnL** did not exhibit d-d bands, but only exhibited π-π and n-π* bands due to organic ligands around 270 and 380 nm, respectively. On the other hand, an intense HSA peak appeared around 300 nm, attributed to n-π* bands of peptides, and a relatively weak **ZnL** peak of 363 nm disappeared. After docking to form HSA+**ZnL**, a spectral change of the latter peak could be observed predominantly.

#### 2.1.2. CD Spectra

Figure 2a,b depict CD spectra (in DMSO) of HSA and HSA+**ZnL**, respectively. There were several CD peaks of HSA in the region shown, ascribed to chirality of *L*-amino acids components, while hybrid material appeared as a broad peak only. A clear difference constitutes proof of formation of the hybrid material of HSA+**ZnL**. A possibility also exists that the secondary structure of HSA would be changed by inclusion of **ZnL**, although characteristic features of secondary structures result in a change of CD spectra at shorter wavelength regions, generally [31]. On the contrary, structural change of **ZnL** by the surrounding environment of HSA, if possible, could neither be observed nor discussed based on Figure 2.

#### 2.1.3. Fluorescence Spectra

In this section, appropriate wavelengths of optical and fluorescence spectra lead to FRET, which suggests short distance approaching of the guest **ZnL** molecules. Figure 3 depicts fluorescence (PL) spectra (in DMSO) of **ZnL**, HSA, and HSA+**ZnL**. According to Figure 3a,c, the optimal condition for emission was determined to be **ZnL** (λ_ex_ = 360 nm, λ_em_ = 421 nm) and HSA (λ_ex_ = 286 nm, λ_em_ = 349 nm). However, **ZnL** can also emit under the same condition (λ_ex_ = 284 nm, λ_em_ = 421 nm), which is the same excitation and different emission as HSA (Figure 3b, marked as broken circles). Therefore, the hybrid material HSA+**ZnL** (λ_ex_ = 285 nm) was measured, and an intense emission band at 421 nm could consequently be observed, as shown in Figure 3d. 

Fluorescence resonance energy transfer (FRET) could also be exhibited, shown as pink arrows in Figure 3d, because the emission wavelength of HSA (349 nm, marked as an orange circle) is close to the excitation wavelength of **ZnL** (360 nm, marked as a pink circle). This is one of the conditions in which FRET occurs generally [39]. Moreover, observation of FRET suggests the distance between donor (HSA) and acceptor (**ZnL**) may be within 1.0 nm. This circumstance can also be supported by two-dimensional (2D) contour plots (Figure 4). Commonly, structural features of HSA dominate intermolecular interaction towards other molecules [40,41].

### 2.2. DFT Calculations of Zn(II) Complexes

#### 2.2.1. Geometry Optimization in the Isolated State

In this section, both calculated and measured IR spectra are mentioned in order to know the certain wavelengths of infrared light which can be absorbed by **ZnL** molecules. This molecule appears optimized as a symmetric structure through the Zn···C36 plane, although both fragments are remarkably rotated, as shown in Figure 5. The C3-O25-Zn-O torsional angle has a value at the B3LYP/6-31G(d,p) level of −128.5º, which is close to that obtained by MP2/6-31G(d,p), −127.1º. 

Because the negative charge on the oxygen atom is higher than that on the nitrogen atom, the O–Zn bond length is shorter than the N–Zn, 1.894 Å vs. 2.019 Å, respectively, by MP2. A slight difference appears in B3LYP, as shown in Table 1. This negative charge, which is higher on the oxygen atoms than on the nitrogen atoms (Table 2), also leads to a higher repulsion of both oxygen atoms (O···O = 3.276 Å) than both nitrogen atoms (N···N = 2.800 Å), to a higher value of O–Zn–O than N–Zn–N angle, and consequent asymmetry around the Zn atom. The value of this negative charge on the oxygen atoms is similar to that calculated by us on the oxygen atoms O(Zn) in the related Zn(II) complex molecules of ZnAHN and ZnVHN (ca. −0.80*e*^-^ by B3LYP), and in the dipeptides ZnGlyGlyH and ZnGlyGlyOH (ca. −0.77*e*^-^) [31] but it is noticeable higher than that calculated in the uracil molecule on the carbonyl oxygens O2 (−0.619*e*^-^ by B3LYP) and O4 (−0.586*e*^-^) and on related molecules. Because of the high value of this negative charge on the oxygen atoms, it is expected that the estabilization of these ZnL complexes inside of the HSA protein molecules is through the formation of intermolecular H-bonds/interactions with amide groups of HSA. The low value of the charge on the hydrophobic benzene rings of ZnL complexes facilitates that complexes appear to be circled by the hydrophobic region of the HSA molecules, as we found by docking calculations (Section 2.3.2). The negative charge on the nitrogen atoms is slightly lower than that calculated in dipeptides complexes on Zn(II) [31] (ca. −0.635*e*^-^) and on uracil (−0.66*e*^-^), which leads to lower interactions through these atoms and to a slightly longer C=N bond (1.297 Å in ZnL vs. ca. 1.290 Å in ZnGlyGlyH), i.e., the ν(C=N) stretching mode appears in ZnL at a slightly lower wavenumber (experimentally by IR in ZnL at 1647 cm^−1^ vs. 1651 cm^−1^ in ZnGlyGlyH). 

The oxygen atom is almost in-plane with the benzene ring, with a value of the torsional, angle C1–C2–C3–O = 179.4º by MP2. However, as is expected, the nitrogen atom is slightly out-of-plane, with a C5–C4–C9–N = −174.9º (C3–C4–C9–N28 = 9.0º) by MP2. The Zn atom also appears slightly out-of-plane related to the benzene ring, with a value of the torsional angle C2–C3–O25–Zn = 169.0º. These features lead to a rotation of both fragments of the molecule.

#### 2.2.2. Molecular Properties from DFT

Calculated wavenumbers were employed to yield thermodynamic properties, which are presented in Table 3. The theoretical data can be utilized to correct experimental thermochemical information at 0 K, as well as for the effect of the zero-point vibrational energy (ZPVE).

To observe the convergence of the energy and thus to analyze the quality of the theoretical results, several basis sets were used. A small increase of energy was observed with the increment of the 6-31G(d,p) basis. Therefore, the results obtained in these tables on this basis can be considered acceptable. Several thermodynamic parameters, such as enthalpy, heat capacity, free energy, and entropy were calculated in **ZnL**. Small differences in the parameters were observed with the increase of the basis set. The entropies calculated for several kinds of compounds and at different ab initio levels have been reported [42] to have mean absolute deviations of less than 5%, as compared to the experimental data. The differences have been ascribed to the neglect of residual (orientation) entropy present at 0 K in the crystal.

#### 2.2.3. Scaling the Wavenumbers

To improve the computed wavenumbers, one of the most effective procedures of scaling [43,44,45,46] was used, i.e., the two linear scale equation procedure, TLSE. With this procedure of scaling, the error obtained in the scaled wavenumbers is, in general, lower than 5%, which permits an accurate correlation with the experimental bands and thus their assignments. This procedure of scaling requires the previously calculated wavenumbers of model molecules determined at the same computational level. The procedure was developed by one of the authors and represents a compromise between accuracy and simplicity, and thus it was the only procedure used in Figure 6 and Figure 7 and in Table 4 for scaling the wavenumbers.

#### 2.2.4. Vibrational Wavenumbers

The spectra of **ZnL** are divided into two regions: 3700–2000 cm^−1^ and 1900–400 cm^−1^, according to the experimental one obtained. The scaled IR spectra were simulated using TLSE [44,45], and they were compared to the experimental FT-IR spectra, as shown in Table 4. Only some of the more characteristic values are presented in this Table (the full table is available as Appendix A (Appendix A)). 

The first column lists the calculated harmonic wavenumbers (ν^cal^, cm^−1^) in increasing order, with their absolute (A) and relative (A%) infrared intensities in the second and third columns, respectively. The relative intensities were obtained by normalizing the computed value to the intensity of the strongest band. The reduce masses (µ) and force constants (f, mDyne/Å) of each calculated wavenumber are included in the fourth and fifth columns, respectively. The scaled wavenumbers (ν^scal^, cm^−1^) using the TLSE procedure are listed in the sixth column. These values can be directly compared to the experimental wavenumbers by IR (ν^exp^), which are presented in the seventh column. Finally, the eight column shows the calculated percentage potential energy distribution (PED) of the different modes for each computed wavenumber. Contributions lower than 10% were not considered. The Wilson notation was used for the characterization of the benzene ring normal modes [43].

Figure 6 and Figure 7 show the experimental and scaled IR spectra with the wavenumber of the main bands. The main discrepancy between the experimental and scaled spectra corresponds to the observed bands with strong IR intensity at 2364 and 2356 cm^−1^, and with medium intensity at 2341, 2336, and 2326 cm^−1^. All of these bands appear in the intermolecular H-bond region, which is presented in the crystal. Because several bands appear and exhibit strong intensity, strong H-bonds/interactions could be presented in the crystal, but this was not simulated in our simplified isolated state model, or in our dimer system (Figure 6). Because an X-ray study has not yet been reported, we cannot calculate these interactions sufficiently well to improve our model. Another possible interpretation is that they correspond to the stretching ν(CO_2_) gas at 2350 cm^−1^ which was not compensated. There also appears a band at 671 cm^−1^ which is also characteristic of the CO_2_ group.

A broad experimental band with medium intensity at 1738 cm^−1^ is also noted, which was not predicted in our calculations. Because of the position of this band, it could correspond to ν(CO) with a noticeable short C–O bond length as a consequence of H-bonds in the crystal. 

### 2.3. Damage of HSA (+Zn(II) Complex) by IR-FEL

#### 2.3.1. Conditions for IR-FEL Irradiation

Here, the results of IR-FEL irradiation are presented in order to obtain the changes of the secondary structures of **HSA** molecules damaged. The secondary structure analysis, which can deal with not amounts but ratios of peptide chains, was derived from IR spectra of **HSA** after IR-FEL irradiation. First, to evaluate the irradiation effects of the cast films, we adopted FT-IR [47] and infrared microscopy under reflection mode. The former is suitable for the spectral survey of the whole sample, whereas the latter is essential for the measurement of spectral in a restricted area of the surface. To measure the spectrum at the same place, an X mark is scratched on the cast film to make a reference point. A typical image of the sample surface is shown in Figure 8, in which black areas representing the cast film of HSA and gray areas corresponding to the phosphoric buffer can be recognized. By careful adjustment of the square cursor on the black area, as indicated in Figure 8, we can obtain spectral information on the local area of the HSA film.

Second, we utilized protein analysis software (IR-SSE) [48] in order to statistically estimate the ratio of α-helix, β-sheet, and β-turn by analysis of the spectral shape around 1620 cm^−1^. 

Third, to decide the wavelength of IR-FEL, we measured FT-IR spectra of HSA and HSA**+ZnL**, as shown in Figure 9. Vibrational analysis for the **ZnL** spectra was performed by the combination of TD-DFT, B3LYP, and 6-31 G (d, p). Selected irradiation wavelengths in the present study were amide I (1652 cm^−1^ = 6.05 μm) and amide II (1537 cm^−1^ = 6.50 μm) corresponding to the HSA intense peaks [49], and C=N (1622 cm^−1^ = 6.17 μm) corresponding to the **ZnL** intense peak. 

Finally, we confirmed that no change was seen for the FT-IR spectra of **ZnL** embedded in KBr before and after IR-FEL irradiation for 30 min at the wavelength of C=N (1622 cm^−1^), which indicated that **ZnL** could not be dissociated by IR-FEL under the power employed in the current experiment.

#### 2.3.2. Changes of HSA and HSA+ZnL after IR-FEL Irradiation

Figure 10, Figure 11 and Figure 12 exhibit FT-IR spectra of the cast films of HSA+**ZnL** and HSA before and after the irradiation of IR-FEL at 1622 cm^−1^ (C=N), 1652 cm^−1^ (amide I), and 1537 cm^−1^ (amide II), respectively. As shown in Figure 10a,b, no significant change was observed for the spectral contour of amide I band in both films after irradiation times of 5, 10, 20, and 30 min. This fact suggests that, although the molecular structure of HSA was changed by docking with **ZnL**, HSA did not dissociate by IR-FEL irradiation at 1622 cm^−1^ because of the low absorbance of HSA at this wavelength. Furthermore, from the results of IR-SSE, little change was found for the ratio of protein secondary structure. Accordingly, the docking of **ZnL** did not affect the dissociation of films at this wavelength. In the case of 1652 and 1537 cm^−1^, the amide I peak shape of HSA+**ZnL** changed more markedly than HSA alone with the irradiation time. Moreover, with IR-SSE, the ratio of protein secondary structure of HSA+**ZnL** changed more notably than did HSA alone. Therefore, **ZnL** docking into HSA seems to promote dissociation by IR-FEL, the reason for which is investigated in the following sections. Docking simulation of HSA+**ZnL** with a GOLD program [50], through several problems gave neither deterministic information nor reliable results about detailed docking sites of **ZnL** in the HSA molecule unfortunately.

When IR-FEL was irradiated to the amide I and amide II bands of HSA+**ZnL**, the content of the α-helix significantly changed compared to HSA alone. Unfortunately, because DFT calculations about protein molecules, as well as hybrid materials of protein-metal complexes, may be difficult, vibrational analysis, which requires hybrid materials for proving this finding, would also be impossible, even in the near future. Contrary to the effect induced by UV-Vis light (electronic excitation), IR irradiation (vibrational excitation by IR-FEL) made the composite of HSA+**ZnL** more unstable than HSA only.

## 3. Materials and Methods

### 3.1. General Procedures

Chemicals of the highest commercial grade available and HSA (F-V) (Nacalai Tesque, Kyoto, Japan) were purchased from Aldrich (St. Louis, MO,USA), Wako (Osaka, Japan) and TCI (Tokyo, Japan), and used as received without further purification.

### 3.2. Preparations

**ZnL** was prepared according to the literature method [51], and confirmed with predominant peaks of IR (1638 cm^−1^, C=N) and UV-vis (268 and 363 nm) spectra. 

Commercially available HSA (1.67 × 10^−7^ mol/L) and 1-propanol solution of **ZnL** (4.9 × 10^−4^ mol/L) were mixed in sodium citrate buffer (2 mL), dropped onto a Parafilm, and dried for approximately 2 days at 298 K to give rise to a cast film. We also prepared a cast film of HSA only.

### 3.3. Physical Measurements

Infrared (IR) spectra were recorded by transmission mode, using KBr pellets for **ZnL** only and by reflection mode as cast films for HSA and HSA+**ZnL** using a stainless plate on a JASCO (Tokyo, Japan) FT-IR 4200 plus spectrophotometer in the range 4000–400 cm^−1^ at 298 K. Electronic (UV-Vis) spectra were obtained on a JASCO (Tokyo, Japan) V-570 UV-vis-NIR spectrophotometer in the range 1500–200 nm at 298 K. Fluorescence spectra were measured on a JASCO (Tokyo, Japan) FP-6200 spectrophotometer in the range of 720–220 nm. Circular dichroism (CD) spectra were obtained on a JASCO (Tokyo, Japan) J-820 spectropolarimeter in the range 900–250 nm at 298 K.

### 3.4. Computational Methods

The calculations were carried out by using the MP2 ab initio method and by employing density functional methods (DFT) [52], including the Becke’s three-parameter exchange functional (B3) [53] in combination with the correlational functional of Lee, Yang, and Parr (LYP) [54]. The B3LYP constitutes the most cost-effective method [55,56], and thus it was the only one used in the present manuscript. The B3LYP method was chosen because different studies demonstrated that the data obtained with this level of theory were in good agreement with those obtained in other more computationally-costly methods, such as MP2 calculations, and it predicts vibrational wavenumbers of DNA bases better than the HF and MP2 methods [57]. These methods appear to be implemented in the GAUSSIAN 09 program package [58]. The UNIX version with standard parameters of this package was run in the alpha computer of the University Complutense of Madrid.

Several basis sets were used, starting from the 6-31G(d,p) to 6-311++G(3df,pd). It was noted that 6-31G(d, p) leads to results that represent a compromise between accuracy and computational cost [43,44,45,46], and thus the results were only presented with this basis set. Although a low basis set or a poor (approximate) DFT method for a system such as ours can lead to erroneous conclusions, all of our calculations were performed with the accurate B3LYP method and a large basis set.

The optimum geometry was determined by minimizing the energy, with respect to all geometrical parameters without imposing molecular symmetry constraints. Berny optimization under the TIGHT convergence criterion was used. The keyword FREQ was employed for the wavenumber calculations in harmonic approximation. No imaginary wavenumber was present in the DFT calculated spectra. The natural NBO atomic charges [59,60] are currently one of the most accurate ways to correlate properties and, for this reason, they were the only ones studied in detail. They were determined with the keyword POP=NPA. It should be emphasized that the net atomic charges obtained from Mulliken population analysis show extremely strong basis set dependence, and they change considerably with the method employed in calculations. Therefore, they were omitted in the present manuscript. 

A complex and protein docking simulation was attempted using GOLD suite calculation software (ver. 5.5.0) (CCDC, Cambridge, UK) using HSA (1BM0) from Protein Data Bank [17].

### 3.5. IR-FEL Irradiation and Analysis

Samples of **ZnL** as KBr pellets and cast films of HSA and HSA+**ZnL** composite were prepared for IR measurements. The formation of composites was confirmed by a spectral change of UV-Vis and CD spectra. IR-FEL was used at the Infrared-Free Electron Laser Research Center of Tokyo University of Science (FEL-TUS) [61]. Three wavelengths of IR-FEL irradiation were determined as C=N double bond band 6.17 μm (1622 cm^−1^), amide I band 6.05 μm (1652 cm⁻^1^), and amide II band 6.50 μm (1537 cm⁻^1^).The iIntensity of the IR-FEL beam was appropriately tuned not to break the chemical bonds in **ZnL** for 30 min irradiation, which was confirmed with IR spectra prior to subsequent experiments using HSA. By comparing the IR spectra before and after irradiation, the changes in each structure (α-helix and β-sheet etc.) were quantified and protein secondary structure analysis was performed using analytical software IR-SSE (Jasco Co., Japan) [48].

## 4. Conclusions

In this study, we revealed the docking of **ZnL** into HSA by using UV-Vis, PL, CD, and docking simulation with GOLD. At the wavelengths of 1622 cm^−1^ (C=N absorb peak), 1652 cm^−1^ (amide I), and 1537 cm^−1^ (amide II), IR-FEL light was irradiated to the cast films of HSA and HSA+**ZnL**. Changes of the samples were evaluated, and dissociation of these films was estimated by FT-IR with IRM and IR-SSA. From these results, it was demonstrated that HSA+**ZnL** is more dissociated than HSA solely when IR-FEL is irradiated at the wavelength of amide I and amide II of HSA. These results could provide useful insights for future practical applications of salen-type Schiff base metal complexes. It can be seen that the complexation of the complex accelerated the degradation rather than alleviating the degradation due to the absorption of infrared rays, so that the stable three-dimensional structure of the protein was disturbed and the protein was easily degraded. However, at present, the exact binding mechanism (for example, an experimental crystallographic study) between the complex and the protein is unknown. Comparisons of different metal complexes and proteins are ongoing.

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
