# Peer review of "Degradation of Human Serum Albumin by Infrared Free Electron Laser Enhanced by Inclusion of a Salen-Type Schiff Base Zn (II) Complex"

_ijms, 2020, doi:10.3390/ijms21030874_

Round 1

Reviewer 1 Report

In the manuscript, authors studied with different techniques (UV-Vis, CD, PL, DFT) the interaction between a Zn(II) complex and the human serum albumin (HSA) protein. The aim is very interesting and several techniques are used, but the main problem is the absence of a common thread. In particular, the results and discussion section seems only a long list of the techniques and relative results. There are not efforts for a common discussion and to explain how the different results are linked to each other. Indeed, also conclusions are a list of results of different techniques. Before publication, I suggest a strong revision to improve the homogeneity of the results and discussion section as indicate. Moreover, other major revisions are list below:

1) In Table 2, authors reported the atomic charges, but in the manuscript is not explained which kind of useful information we can derived from these values.

2) At pg.9, authors suggested that the bands around 2360 cm-1 are due to the "H-bonds/interactions through the oxygen atoms and the hydrogen of the CH2 groups". It is possible to verified this hypothesis with the inclusion of a single explicit solvent molecule at different position. The system is small enough to allow these additional calculations. Without them, more than half spectrum in Fig 6 is not reproduced.

3) The reported formula seems not correct. The DeltaG bind should be the difference between the HSA+ZnL-ZnL-HSA. Moreover, not additional explanations are reported for DeltaGbind in the manuscript.

4) All the computational setup for GOLD program (i.e. Value, temperature, number of solvent molecules,...) is missing.

Minor revisions:

1) The x axe scale of Figure 1 is not appropriate. It will be sufficient to arrive at 500 nm to improve the clarity of the relevant part (between 200 and 400 nm)

2) In figure 5, the grey atom labels are not clear. A different color must be used

3) The dimension of the first column in Table 1 has to be adjust to have in the same line the atom labels.

4) Figure 7 is not well reproduced.

5) The resolution of figures 10-12 is very poor.

Author Response

<Reviewer 1> 1) In Table 2, authors reported the atomic charges, but in the manuscript is not explained which kind of useful information we can derived from these values. @According to your comments, we have mentioned explanation as follows: “(Table 2), also leads to a higher repulsion of both oxygen atoms (O···O = 3.276 Å) than both nitrogen atoms (N···N = 2.800 Å), a higher value of O-Zn-O than N-Zn-N angle, and consequent asymmetry around the Zn atom.”

2) At pg.9, authors suggested that the bands around 2360 cm-1 are due to the "Hbonds/interactions through the oxygen atoms and the hydrogen of the CH2 groups". It is possible to verified this hypothesis with the inclusion of a single explicit solvent molecule at different position. The system is small enough to allow these additional calculations. Without them, more than half spectrum in Fig 6 is not reproduced. @According to your comments, we have been doing calculations according to point 2 of the reviewer. In the DFT calculations carried out, we have not found a wavenumber close to 2360 cm-1. So the probably answer is that when the spectrum was recorded, the CO2 of the air was not compensated previously.

3) The reported formula seems not correct. The DeltaG bind should be the difference between the HSA+ZnL-ZnL-HSA. Moreover, not additional explanations are reported for DeltaGbind in the manuscript. @ According to your comments, we have removed the formula about delta G in Figure 13. Moreover, the delta G values used here are usually employed in the GOLD program.

4) All the computational setup for GOLD program (i.e. Value, temperature, number of solvent molecules,...) is missing. @ According to your comments, we have added important setups for a GOLD program.

Minor revisions:

1) The x axe scale of Figure 1 is not appropriate. It will be sufficient to arrive at 500 nm to

2

improve the clarity of the relevant part (between 200 and 400 nm) @ According to your comments, we have improved Figure 1.

2) In figure 5, the grey atom labels are not clear. A different color must be used @ According to your comments, we have improved the color of Figure 5.

3) The dimension of the first column in Table 1 has to be adjust to have in the same line the atom labels. @ According to your comments, we have improved Table 1.

4) Figure 7 is not well reproduced. @ According to your comments, we have improved Figure 7.

5) The resolution of figures 10-12 is very poor. @ According to your comments, we have improved the resolution of Figures 10-12.

Reviewer 2 Report

The article by Onami and colleagues is very difficult to read, given its problems in English usage. I suggest engaging a language consultant. For instance, in the abstract, the sentence 

The wavelength of IR-FEL irradiation was determined based on experimental FT-IR spectra and vibrational analysis.

falsely suggests that the authors used FT-IR spectra and vibrational analysis in order to measure the wavelengths used to irradiate the protein. That makes no sense to me. I believe that the authors’ intent is better indicated when the words to be used are inserted, and the sentence is restated as

The optimum wavelengths of IR-FEL irradiation to be used were based on experimental FT-IR spectra and vibrational analysis.

The article is full of problematic sentences. If there were only a few of them, then I would correct them, but that work is beyond the scope of this review.

The experimental setup for IR and other spectra seems fine, insofar as I can decipher the text. However, the docking experiment is presented in a way that provides no information at all. Figure 13, which purports to show the docked ligand, is totally useless for the purpose. The authors should attempt to provide an image of a docked ligand that shows what amino acids interact with the ligand. Those amino acids must be identified individually. The PDB database itself provides examples of such images, and the use of a specialized display program, such as the ligand map feature in Molegro Viewer would be very helpful.

It would be informative if the authors compared their ZnL docking results with the binding of the many other human serum albumin ligands for which structures are known. Does ZnL go into the same binding pocket or pockets as do any of the other known ligands?

Most important are the following two points.

What is the binding affinity of ZnL to HSA? The authors ought to provide a dissociation constant measured by a direct experimental method, not by predictive docking. The authors’ spectral results could be used for this purpose, perhaps, or one could use any number of other methods, including simple dialysis. In the conclusions section, the authors state that HSA + ZnL is more dissociated than HSA solely when IR-FEL is irradiatedThis sentence suggests that IR-FEL is an object of some type that is irradiated in these experiments, which is clearly not the authors’ meaning. Getting past that problem of language, the authors ought to report the binding affinity (as a dissociation constant) of ZnL to HSA after irradiation. If irradiation makes for a more dissociated state, i.e. less binding, then the authors can report on this modified binding in a quantitative way. 

Author Response

The article by Onami and colleagues is very difficult to read, given its problems in English usage. I suggest engaging a language consultant. For instance, in the abstract, the sentence

The wavelength of IR-FEL irradiation was determined based on experimental FT-IR spectra and vibrational analysis.

falsely suggests that the authors used FT-IR spectra and vibrational analysis in order to measure the wavelengths used to irradiate the protein. That makes no sense to me. I believe that the authors’ intent is better indicated when the words to be used are inserted, and the sentence is restated as

The optimum wavelengths of IR-FEL irradiation to be used were based on experimental FT-IR spectra and vibrational analysis.

The article is full of problematic sentences. If there were only a few of them, then I would correct them, but that work is beyond the scope of this review.

@According to your comments, we have fixed the sentences. As for English usage, English correction has been already carried out by a professional person in USA before the first submission.

3

The experimental setup for IR and other spectra seems fine, insofar as I can decipher the text. However, the docking experiment is presented in a way that provides no information at all. Figure 13, which purports to show the docked ligand, is totally useless for the purpose. The authors should attempt to provide an image of a docked ligand that shows what amino acids interact with the ligand. Those amino acids must be identified individually. The PDB database itself provides examples of such images, and the use of a specialized display program, such as the ligand map feature in Molegro Viewer would be very helpful. It would be informative if the authors compared their ZnL docking results with the binding of the many other human serum albumin ligands for which structures are known. Does ZnL go into the same binding pocket or pockets as do any of the other known ligands? @ Of course, I would like to agree with our comments was right in common cases. The purpose of this study is comparison of docking to the same protein (HSA) with ZnL and other (peptide-based) Schiff base Zn complexes in our previous paper [28], in which the issues due to conformational changes of the complexes was stated clearly. Hence, we limited the description of docking in the qualitative level in this study.

Most important are the following two points. What is the binding affinity of ZnL to HSA? The authors ought to provide a dissociation constant measured by a direct experimental method, not by predictive docking. The authors’ spectral results could be used for this purpose, perhaps, or one could use any number of other methods, including simple dialysis. In the conclusions section, the authors state that HSA + ZnL is more dissociated than HSA solely when IR-FEL is irradiated … This sentence suggests that IR-FEL is an object of some type that is irradiated in these experiments, which is clearly not the authors’ meaning. Getting past that problem of language, the authors ought to report the binding affinity (as a dissociation constant) of ZnL to HSA after irradiation. If irradiation makes for a more dissociated state, i.e. less binding, then the authors can report on this modified binding in a quantitative way. @We have added the following explanation in the conclusion section. “It can be seen that the complexation of the complex accelerated the degradation rather than alleviating the degradation due to the absorption of infrared rays, so that the stable three-dimensional structure of the protein was disturbed and the protein was easily degraded. However, at present, the exact (for example, experimentally crystallographic study) binding mechanism between the complex and the protein is unknown.”

Round 2

Reviewer 1 Report

The authors improved the quality of the manuscript and they answered to all my previous queries, but the main problem that I highlighted remains: the absence of a common thread. Again, the manuscript seems a  long list of the techniques and relative results. Again, I suggest a revision to improve the homogeneity of the results and discussion section before publication.

Author Response

2020 January 22

Dear The editor of International Journal of Molecular Science:

Please consider the following reply for query and THE SECOND revised manuscript (ijms-691470) uploaded. Correction were highlighted or red letters in the revised manuscript.

<Reviewer 1>

The authors improved the quality of the manuscript and they answered to all my previous queries, but the main problem that I highlighted remains: the absence of a common thread. Again, the manuscript seems a long list of the techniques and relative results. Again, I suggest a revision to improve the homogeneity of the results and discussion section before publication.

@According to your comments, we have mentioned whole explanation and relationship of each items before “results and discussion” (at the end of introduction) and the beginning of each section in “results and discussion” for homogeneity that “start sentences are explanation”.

<Reviewer 2>

The significance of the interactions described by the authors can only be evaluated if the authors present information on the location of binding by their ligands to the protein used in the study. The authors fail to do this. They even state that the location of binding is unknown, even though docking provides this information.

If the authors are having technical problems in using Gold, nothing prevents them from using other software, such as Autodock Vina.

In response to this comment in the previous round of review, they have added a reference to a prior paper, but that paper also fails to provide that information. It is paradoxical that, if the authors actually have performed the in-silico docking with Gold, it would be simple for them to answer this criticism and provide the image or list of the protein - ligand interactions. See page 122 and 123 of the Gold user guide.

The language in the docking section is strange, for instance, where I see the sentence

Additionally, folding of peptide around the including site of ZnL 311 became loose without retaining the original structure.

It is unclear how the authors know that the peptide folding became loose. How was that determined? NMR?

It would be interesting to know if their ligand is predicted to bind in the same place as other ligands, such as lidocaine in the 3JQZ pdb structure. Their Gold docking results must answer that question.

Also, the authors continue to fail to present any data on the experimental dissociation constant for their ligand to HSA. While referring frequently to the delta G of interaction, they fail to give any predicted or experimental values for the delta G of binding from which a molar dissociation constant could be calculated.

@ We examined the results of docking with GOLD, though some points could not be improved nor overcome at all. Therefore, we omitted the part about docking with GOLD from the revised manuscript, which did not change the results of this study. However, docking of metal complex into protein were exhibited by fluorescence spectra experimentally. We have added this sentence in 2.3.1, “Docking simulation of HSA+ZnL with a GOLD program [46], though several problems did not give deterministic and reliable results about detailed docking sites of ZnL in HSA molecule unfortunately.”.

That’s all.

Best regards,

Takashiro Akitsu

Department of Chemistry, Faculty of Science, Tokyo University of Science

1-3 Kagurazaka, Shinjuku-ku, Tokyo 162-8601, Japan

Tel. +81-3-5228-8271 (ext. 5775)

Fax. +81-3-5261-4631

E-mail. <[email protected]>

Reviewer 2 Report

The significance of the interactions described by the authors can only be evaluated if the authors present information on the location of binding by their ligands to the protein used in the study. The authors fail to do this. They even state that the location of binding is unknown, even though docking provides this information. 

If the authors are having technical problems in using Gold, nothing prevents them from using other software, such as Autodock Vina. 

In response to this comment in the previous round of review, they have added a reference to a prior paper, but that paper also fails to provide that information. It is paradoxical that, if the authors actually have performed the in-silico docking with Gold, it would be simple for them to answer this criticism and provide the image or list of the protein - ligand interactions. See page 122 and 123 of the Gold user guide. 

The language in the docking section is strange, for instance, where I see the sentence

Additionally, folding of peptide around the including site of ZnL 311 became loose without retaining the original structure.

It is unclear how the authors know that the peptide folding became loose. How was that determined? NMR? 

It would be interesting to know if their ligand is predicted to bind in the same place as other ligands, such as lidocaine in the 3JQZ pdb structure. Their Gold docking results must answer that question. 

Also, the authors continue to fail to present any data on the experimental dissociation constant for their ligand to HSA. While referring frequently to the delta G of interaction, they fail to give any predicted or experimental values for the delta G of binding from which a molar dissociation constant could be calculated.

Author Response

(The authors gave the same response as above.)

Round 3

Reviewer 2 Report

The understandability of the paper has been much improved since the last version. 

Requests to provide detailed output from the authors' work using the Gold docking program have been met by removal of the inadequate output (and inadequate images) that were present in the original manuscript. 

I see no meaningful response to the request to provide an experimentally determined dissociation constant for the ligand to the protein, perhaps by using dialysis (although other methods would also suffice.) This requires a method separate from the spectroscopic data.

I also see no dissociation constant (in moles/liter) that is derived from their spectroscopic data, one that can be replicated by other researchers.

Since the authors continue to decline to give output data from the Gold docking calculations, then all reference to that docking work should be removed from the paper and the abstract. The words Gold and docking should not appear anywhere.   

If the authors decline to give a dissociation constant for their ligand to the protein, then the experimental reasons for that failure should be addressed in the manuscript.